# Evaluating Management Strategies for Mount Kenya Forest Reserve and National Park to Reduce Fire Danger and Address Interests of Various Stakeholders

**Kevin W. Nyongesa \* and Harald Vacik**

Institute of Silviculture, University of Natural Resources and Life Sciences, Vienna (BOKU),
Peter-Jordan-Strasse 82, A-1190 Vienna, Austria; harald.vacik@boku.ac.at
**\*** Correspondence: kevin.nyongesa@boku.ac.at; Tel.: +4368860599334

**Abstract:** A Multi-Criteria Analysis (MCA) approach was employed for evaluating and selecting the best management strategy for Mount Kenya Forest Reserve and National Park (MKFRNP). The MCA approach used a set of objectives and criteria (O&C) to address the complexity of the decision problem in a transparent and understandable way, which also facilitated the active participation by diverse professionals, experts, and interest groups. The management strategies were developed to fulfill the key components of MKFRNP management and the current situation in the study area. The seven management strategies focused on climate change mitigation, protection of water catchments, education and research, stakeholder involvement, biodiversity conservation, timber production, and community interests. Forest stations with differing fire danger levels (very high, high, moderate, and low) were selected to compare the performance of the management strategies. The strategies were assessed qualitatively on their potential to improve the current situation according to the entire set of O&C. The Analytic Hierarchy Process (AHP) was employed to identify the best management strategy according to the overall preferences of all stakeholder groups. The AHP indicated that a strategy focusing on community interests provided the best option to address the current management challenges in all the seven forest stations independently of their fire danger levels. Biodiversity conservation should also be considered by resource managers in order to reduce fire danger and increase the benefits obtained by different stakeholders in MKFRNP.

**Keywords:** multi-criteria analysis (MCA); objectives and criteria (O&C); fire danger; benefits; forest managers; wildlife managers; management strategies (MS); community forest associations (CFAs); Stakeholders; analytic hierarchy process (AHP)

## 1. Introduction

Mount Kenya Forest Reserve and National Park (MKFRNP) is a UNESCO World Heritage Site [1] and a major forested water catchment area in Kenya [2]. It is jointly managed by the Kenya Forest Service (KFS), the Kenya Wildlife Service (KWS), and the Lewa Wildlife Conservancy [3]. Although some management successes for biodiversity conservation have been achieved in the MKFRNP, tremendous threats and pressures still remain [3]. Kenya's fast-growing population has increased pressure on MKFRNP resources over the past three decades [4]. The main source of this pressure arises from the depletion of forest resources and degradation within and near to the populated areas around MKFRNP [1,5,6]. As resources become scarce on private and community land, the population has been turning to the neighboring protected areas of MKFRNP for livelihood resources [7]. This has led to a depletion of resources, degradation, and increased wildfires. Besides human encroachment and land use change, invasive species, allocation of forest land to some communities and influential individuals

by the former governments, cultivation of marijuana (*Cannabis sativa* Linnaeus), pests and diseases, tourist or visitor related impacts such as poor waste management and human-wildlife conflicts have been observed [8]. This is mainly due to unsustainable use levels and patterns that have occurred as a result of poverty, poor or inappropriate management skills and weak management institutions and systems [5].

Natural fires in MKFRNP are caused by lightning [7] but most of the fires are recorded by KFS and KWS as "unknown causes", making it difficult to estimate their social, economic, cultural and ecological effects [7]. According to the KFS and KWS, human-caused fire ignitions in MKFRNP are more likely to increase in the future, because climate change may affect fire season length and severity [4]. Communities living around MKFRNP use fire as a tool in land management and sometimes the fire goes out of control causing unintentional wildfires in MKFRNP [7]. Perennial grassland fires are common in many parts around MKFRNP because each year during the dry season, communities set grasslands on fire to keep them open and to facilitate the growth of new grass for livestock, especially before the rain begins. Farmers around MKFRNP use fire to prepare their farmlands, to break impenetrable bushlands, control weeds, pests, and parasites and to keep wildlife away from homes. Bushland and forest fires are also common in MKFRNP because some community members use fire to burn charcoal, harvest wild honey, and hunt and roast game meat in MKFRNP [4]. Some other activities such as children carelessly playing with fire, the throwing of lit cigarette butts and poor handling of campfires have also contributed to the ignition of wildfires in MKFRNP [7]. Arsonists have caused wildfire ignitions in MKFRNP as a way of revenging on the KFS and KWS for being excluded from accessing some benefits from the forest resources. Wildlife poachers also ignite wildfires to escape being arrested and prosecuted by the KFS and KWS. Inter-community conflicts over water and pasture grounds between the locals (Kikuyu, Meru, and Embu) and the pastoralists (Samburu and Maasai) having been a source of fire ignitions in MKFRNP and are likely to increase [7]. During years of extreme drought, migrant pastoralists usually come to graze in MKFRNP, set fire to the old grass to facilitate the growth of new grass, and then move away in search of good pasture grounds. This practice has been causing huge fires and conflicts, due to the loss of grazing grounds for the locals, primarily the Kikuyu, Meru, and Embu communities who depend on the grasslands within MKFRNP for grazing their livestock [4,7]. Additionally, the intensified cultivation of exotic fire-prone tree species like cypress (*Cupressus lusitanica* Mill.), patula pines (*Pinus patula* Schiede Ex Schltdl. & Cham), radiata pines (*Pinus radiata* D. Don), blue gum (*Eucalyptus saligna* Smith), and rose gum (*Eucalyptus grandis* W. Hill Ex Maiden) will increase fire hazard in the future [4].

Ground fires, surface fires, and crown fires have occurred in MKFRNP grasslands, farmlands, bushlands and forests [4,6]. The KFS and KWS fire records from 1980 to 2017 show that MKFRNP has experienced about 210 wildfires. Most of these wildfires in MKFRNP occurred in the months of January, February, March, September, and October. The fire records also show that from 1980 to 2017, more than 668 Ha of plantations, 21,276 Ha of bushland and grassland, 267 Ha of bamboo, 6727 Ha of indigenous/natural forests and 11,175 Ha of moorland were burned by wildfires in MKFRNP. According to the KFS and KWS fire records from 1980 to 2015, the estimated firefighting cost and the fire damages were $134,759.84 and $4,712,384.96 respectively [6].

The community members and other stakeholders (user groups) with interests in MKFRNP have been obtaining their licenses from the government of Kenya through KFS and KWS [6]. The licenses are mainly obtained for the practices of: conserving biodiversity, conducting education and research, harvesting timber and poles, grazing livestock, collecting firewood, beekeeping, collecting herbs and spices, collecting wild fruits, collecting water, farming trout fish (*Oncorhynchus mykiss* Walbaum), providing hotel and cottage services as well as ecotourism, practicing of cultural rituals, farming under the Plantation Establishment and Livelihood Improvement Scheme (PELIS) and acting as community scouts in MKFRNP [1,3]. Despite the KFS and KWS efforts to license some of the user groups' activities in MKFRNP, not all their interests have been addressed. This is because there have been cases of conflicting interests between the KFS, KWS and the user groups or between the different user groups.

The Mount Kenya Forest Reserve (MKFR) management plan 2010–2019 was developed by the KFS to guide the establishment, development and sustainable management, including conservation and rational utilization of the forest and allied resources for socio-economic development [3]. The MKFR management plan 2010–2019 was prepared in compliance with the legal requirement of the Forests Act, 2005 under section 35 that provides a mandatory legal requirement for preparation of management plans of all state, local authority and provisional forests. The MKFR management plan 2010–2019 considers the draft Forest Policy No. 9 of 2005; the KFS Strategic Plan 2009/10–2013/14; the Environmental Management and Coordination Act (EMCA), 1999; the Wildlife (Conservation and Management) Act, Cap. 376. The MKFR management plan 2010–2019 also considers the Water Act, 2002, other policies and legislative frameworks whose objectives have a direct impact on sustainable conservation, management, and utilization of MKFRNP [3].

Several management strategies that are in line with the MKFR management plan 2010–2019 have been suggested by different stakeholder groups that have interests in MKFRNP. The KFS and saw millers have suggested that a management strategy which satisfies wood production interests, in terms of timber and poles is the best. However, scientific experts from Egerton University, University of Natural Resources and Life Sciences, Vienna (BOKU) and Kenya Forest Research Institute (KEFRI) have suggested that increased plantation establishment for timber and poles will contribute to the national climate change mitigation interests through carbon sequestration and will also improve the protection function such as reduction of soil erosion and landslides. The KFS and Community Forest Associations (CFAs) confirm that an increased timber and poles production will help to achieve income interests through the creation of employment opportunities in the forestry sector [7]. The scientific experts have also suggested the need for establishing plantations with indigenous tree species like Meru oak (*Vitex keniensis* Turrill) and African pencil cedar (*Juniperus procera* Hochstetter. ex Endlicher) that can be used for sawn timber and poles and still contribute to carbon sequestration. Some CFA members have suggested that the best management strategy should satisfy agriculture interests by providing grazing grounds for livestock and using former harvested areas to cultivate crops and plant trees under the Plantation Establishment and Livelihood Improvement Scheme (PELIS) [4]. Additionally, the CFAs suggest that increased access to non-timber forest products such as foraging of wild fruits, hunting of game meat, fishing, collection of honey and herbal medicine would satisfy the needs of the local people. In this context, the energy demands also can be fulfilled by fostering firewood collection and charcoal making [6]. But on the other hand, the scientific experts have suggested that the use of energy efficient cooking stoves by communities can help to reduce the growing demand for firewood collection and charcoal making in MKFRNP. The counties and water resources management authorities (WRMAs) consider that increased charcoal burning and timber harvesting will affect water quality and quantity [7]. They have therefore suggested management strategies that will help to increase the protection of water catchments against human-caused deforestation and degradation [3]. Further, scientific experts, the KWS, biodiversity conservation organizations such as Lewa Wildlife Conservancy (LWC) and Mt. Kenya Wildlife Trust (MKWT) have argued that increased plantations consisting of exotic trees such pines, cypress and eucalyptus will contribute to increased wildfire danger, affect the growth of indigenous trees, decrease wildlife forage and breeding habitats in the MKFRNP [4]. The KWS, LWC, and MKWT suggest that the maintenance of wildlife species diversity, tree species diversity, plant species diversity, the elephant migration corridor, key habitats, and protected areas can meanwhile increase income from tourism. But the scientific experts have suggested that tourist activities in MKFRNP should be well regulated because uncontrolled campfires may cause unintentional wildfires leading to degradation of some key habitats and protected areas. The scientific experts, KWS, LWS, and MKWT also oppose the increase of CFAs access to non-timber forest products, firewood and charcoal because of unsustainable use levels and patterns as a result of poverty that have endangered wildlife and tree species [1].

The Mount Kenya Ecosystem (MKE) management plan 2010–2020 was developed by KWS through a participatory planning process involving various stakeholders, under the coordination

of a core planning team that comprised representatives of KWS and KFS managers and planners, national environmental management authority (NEMA) and the water resources management authority (WRMA) regional officers in charge of the MKE [1]. The MKE management plan 2010–2020 has been developed in line with the KWS Protected Area Planning Framework (PAPF). Unlike other types of plans where management actions are often stated but not expounded, management actions in PAPF-based plans are elaborated to improve understanding increasing prospects of implementation. These plans adopt an ecosystem approach addressing conservation issues holistically and actively involving KWS, KFS, and local communities.

The implementation of the MKFR management plan 2010–2019 and MKE management plan 2010–2020 presents a complex decision-making challenge to the KFS, KWS, community forest associations (CFAs) and other stakeholders on how to reduce fire danger and increase the benefits obtained by different stakeholders in MKFRNP [1,3]. Therefore, there is a strong need to develop and choose the best management strategy for MKFRNP that can help to reduce the fire danger and address the interests of various stakeholder groups [1,3]. Multi-Criteria Analysis (MCA) can help to solve such complex multi-criteria decision problems that include qualitative or quantitative aspects [9,10]. Strong technical and theoretical support for MCA procedures exists, and they are designed to consider an intuitive and transparent participation of multiple experts and stakeholders. Considering stakeholder knowledge will contribute to the general acceptability of the results [9,11,12]. One of the MCA tools that has been used widely by natural resource managers in complex decision-making situations is the Analytical Hierarchy Process (AHP) [13]. The AHP was introduced by Thomas Saaty in 1980 [14] and helps to reduce complex decisions to a series of pairwise comparisons, and then by synthesizing the results, the AHP helps to capture both the subjective and objective aspects of a decision [9]. Therefore, this paper will demonstrate how the AHP can support resource managers, communities and other stakeholders in finding the best management strategy that will help reduce fire danger and increase the benefits obtained by different stakeholders in MKFRNP. The best management strategy will then be jointly implemented by the KFS, KWS, CFAs, WRMAs, NGOs, counties, religious organizations, and other stakeholders with interests in MKFRNP.

The objectives of this project were (i) to develop the Management Strategies (MS) which meet the demands for integrated fire management, (ii) to develop Objectives and Criteria (O&C) for the evaluation of the strategies with all stakeholder groups and (iii) to apply the AHP to propose the best management strategy that reduces fire danger and increases the benefits obtained by various stakeholders in MKFRNP. In the following sections, we will introduce the methodological steps for the MCA, present the case study and draw some conclusions on the selection of the best management strategy.

## 2. Materials and Methods

### 2.1. Description of the Study Sites

MKFRNP is located to the east of the Great Rift Valley, along Latitude 0°10′ S and longitude 37°20′ E. It bestrides the equator in the central highland zones of Kenya [15]. The mountain is situated in two Forest Conservancies and five forest management zones, namely, Nyeri and Kirinyaga in Central Highlands Conservancy and Meru Central, Meru South and Embu in Eastern Conservancy [3]. The MKFRNP has been divided into 23 forest stations as shown in Figure 1. The national park covers 71,510 Ha and the forest reserve covers 213,082.64 Ha [3,16]. Mt. Kenya was formed as a result of volcanic activity and it has a base diameter of approximately 120km. The mountain's highest peaks Batian (5199 m) and Nelion (5188 m) are located in the national park [15]. The altitudes with the highest rainfall are found between 2700 and 3100m, while above 4500m most precipitation falls as snow or hail and frosts are common above 2500m above sea level (asl) [3]. Rainfall pattern in the MKFRNP ecosystem is bimodal. It ranges from 900 mm in the north (leeward side) to 2300 mm on the southeastern slopes (windward side) of the mountain with maximum rains falling during the months of March to June and October to November [16]. The driest months are January and February and the

windward side experiences the strongest effects of the trade wind system. The diurnal temperature ranges in January and February may be as high as 20 °C. The Eastern side (windward side) of the MKFRNP receives more precipitation and is less prone to fires as compared to the Western side (Leeward side) which experiences more fires throughout the year [7]. The climate varies with the altitude and temperatures at MKFRNP are cooler than throughout most of the country. The climate there is either subtropical or temperate. There is still a rainy season from March to May and from October to December when it is drizzly and cloudy [17]. Rainfall is moderate on the lower slopes and heavier higher up. The sunniest months are from December through March [18]. The peak of MKFRNP is always covered in snow [15]. The climate of MKFRNP region is largely determined by altitude. There are great differences in altitude within short distances, which determine a great variation in climate over relatively small distances. Average temperatures decrease by 0.6 °C for each 100m increase in altitude [3]. An afro-alpine type of climate, typical of the tropical East African high mountains, characterizes the higher ranges of MKFRNP [3].

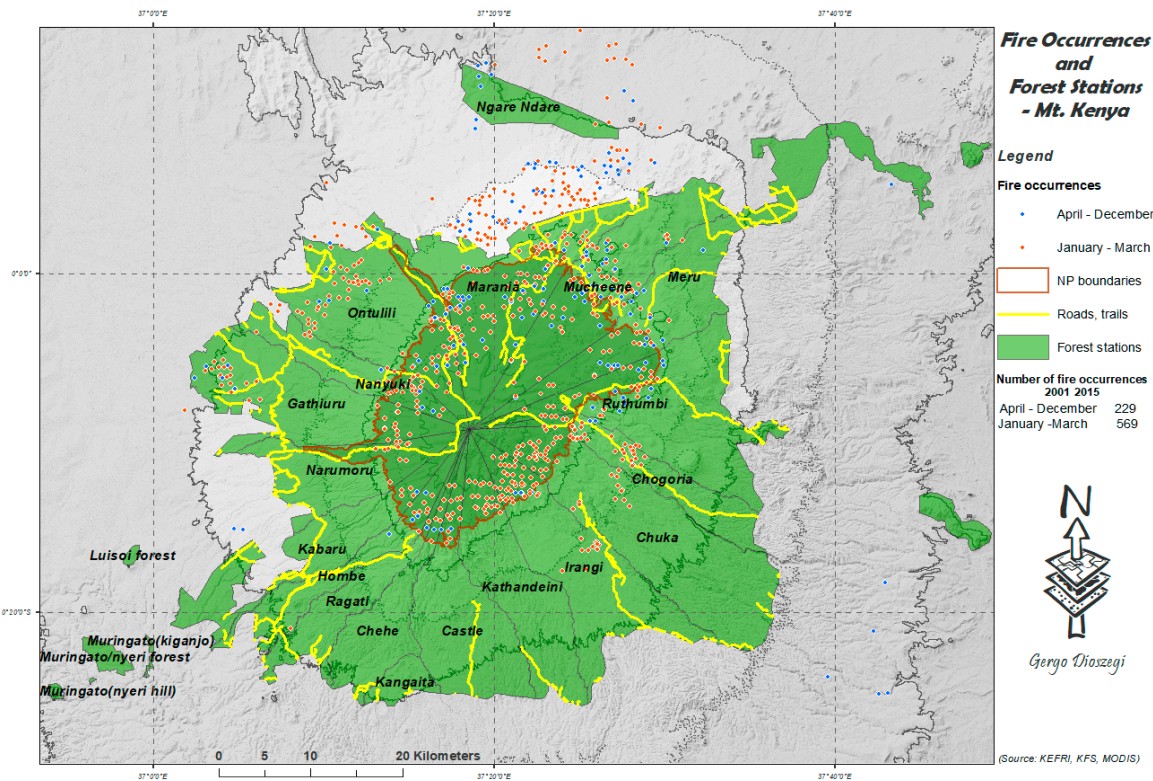

**Figure 1.** A map of Mount Kenya Forest and National Park showing forest stations, national park, roads, trails and the seasonal fire occurrences from 2001 to 2015 (Source: KEFRI, KFS, MODIS).

MKFRNP is an important reservoir for biodiversity and several studies have identified 880 plant species, subspecies and varieties belonging to 479 genera in 146 families below the 3200 m altitude [3]. There are at least 11 endemic species of higher plants and more than 150 species that are near endemic [3]. Vegetation types and species distribution are distinguished according to the different climatic zones and altitudes, most obviously through variation in vegetation structure, cover and composition [3,4]. The moorland (ericaceous belt) lies between 3000 m and 3500 m asl and is mainly covered with giant heath, African sage (*Artemisia afra* Jacquin Ex Willdenow), several gentians (*Swertia spp* Linnaeus), smaller trees in glades, such as the East African rosewood (*Hagenia abyssinica* Willdenow), St. John's wort (*Hypericum spp* Linnaeus) and trees that are covered with moss and lichens (*Usnea spp* Dillenius Ex Adanson) [3]. The pure bamboo (*Arudinaria spp* Michaux) zone occurs between 2550 and 2650m asl while the mixed bamboo with indigenous trees zone extends from 2500 to 3200m asl and is dominated by the African alpine bamboo (*Arudinaria alpina* Schumann), real yellowwood (*Podocarpus latifolius*

Thunberg Ex Mirbel) [3]. The elderberry (*Sambucus africana* Standl.) grows on openings during the transition phase of collapsed bamboo stems [3,4]. The bamboo zone is absent in the northern side of MKFRNP due to drier conditions [3,6]. The indigenous forest zone starts at 2400 m down to 2000 m asl and is dominated by *Podocarpus latifolia* mixed with brittle-wood (*Nuxia congesta* R. Brown Ex Fresen.) at the upper altitudes [3,4,6] while moist forests of East African camphorwood (*Ocotea usambarensis* Engler), forest newtonia (*Newtonia buchananii* Buchanan), woodland croton (*Croton sylvaticus* Hochst. Ex C. Krauss), musine croton (*Croton megalocarpus* Hutchinson), forest bonsai (*Premna bonsai* Linnaeus), silver oak (*Brachylaena huillensis* O. Hoffmann) and cape chestnut (*Calodendrum capense* Thunberg) occur at lower altitudes between 1450 and 2400 m asl [3]. Plantation zone extends from 2200 m to 2400 m asl and the main commercial tree species planted include Cypress, Pines, and Eucalypts while plantations of indigenous species mainly include Meru oak (*Vitex keniensis* Turrill) and African pencil cedar (*Juniperus procera* Hochstetter. ex Endlicher) [4]. Tea (*Camellia sinensis* Linnaeus) is also grown in MKFRNP. The Nyayo Tea Zone was opened up by the Nyayo Tea Zones Development Corporation which was established by Legal Notice No. 265 of 1986 with the aim of promoting forest conservation by providing a buffer zone to check against human encroachment into forest reserves. The total area opened up during the clearing for establishment of the tea belt in Mt. Kenya was 1194.8 hectares out of which 787.9 hectares are currently under tea, 241.8 hectares are under fuel wood plantations and another 165.1 hectares which were also cleared were replanted with indigenous trees [3,4].

Mt. Kenya is considered as a holy mountain according to the Kikuyu community traditions. This is because they traditionally believed that their God "Murungu" or "Ngai" dwelled on top of this mountain. There is also the saying that term Kikuyu originates from the Mukuyu tree (*Ficus sycomorus* Linnaeus) [1,4]. According to the Kikuyu culture, three sacred trees make the community believe that they should conserve the MKFRNP: Mukuyu tree (*Ficus sycomorus* Linnaeus), Mugumo tree (*Ficus thonningii* Blume), and Mukurwe tree (*Albizia gummifera* J.F. Gmel.) [4]. They are used during various rituals and ceremonies [3]. Nobody in the community is allowed to cut down or set fire to these trees and this is similar to other places in Africa and contributes to the efforts of conservation [4].

The Mt. Kenya region has a very high population growth rate [7]. The total population of the communities that live within the districts that border the MKFRNP was 24.4 million in 2009 [3,7]. Human activities in MKFRNP to obtain water, firewood, honey, charcoal, timber, poles, and grass for livestock, income from tourism, game meat, fish, herbal medicine have increased over the past three decades [3,4,6]. Various agro-forestry practices have been adopted which include tree planting in woodlots and cropland around homesteads and along farm boundaries. A good number of people in the area operate small business enterprises that include shops, kiosks, selling milk and other farm products, selling timber and wood products, honey, firewood and charcoal [7]. Other community members engage in quarrying and breaking ballast while others derive their livelihood from providing casual labor on the farms while others are formally employed [1,3,4]. The operating of hotels around the MKFRNP promotes employment both directly and indirectly through the flow of demand for goods and services [4]. A few small-scale farmers around MKFRNP have also initiated fish farming projects for the growing market especially in the local hotel industry [4].

MKFRNP is known to have a long fire history and fire has influenced the vegetation in the landscape because some plant species require fire to germinate, establish, or to reproduce, and total fire suppression not only eliminates these species, but also affects the animals that depend upon them [4,6]. MKFRNP has some fire-dependent species like *Juniperus procera*, *Bambusa vulgaris* (Schrad Ex J.C. Wendl.) and *Hagenia abyssinica* that usually regenerate after fire [4]. Native perennial grasses also regrow from root systems that are rarely damaged by fires that occur in MKFRNP [4,7]. Fire is the only natural factor which also supports the reproduction of the afro-alpine vegetation (chaparral). Older stands of chaparral dry up causing huge fuel accumulation over larger areas thus fire is necessary for the plants to remain vital [4]. However, the current banning of all fires from current land use practices by KFS and KWS might lead to an accumulation of fuel loads, which would play a major role in future wildfire outbreaks in MKFRNP [4].

Plantations of exotic tree species like pines, cypress and eucalypts have been established in MKFRNP by the KFS for the pulp and timber industry. Several studies show that exotic tree species contribute to changes in the patterns of anthropogenic ignitions, flammability of exotic species, forest ecosystem structure, and process and fuel loads [4]. Fire stimulates the release of large amounts of seeds from the serotinous cones of *Pinus radiata* and can create favorable conditions for germination and establishment [4,6,7]. Recovery in fire-resistant Eucalypt species in MKFRNP is by resprouting from epicormic strands (i.e., regeneration from meristem strips, usually extending from the inner to outer bark on aboveground branches and stems, which produce buds), and/or from basal buds. Unmanaged fires may contribute to an increase of exotic species in the natural environment like MKFRNP [4]. However, the use of prescribed fires for fuel management is not practiced by KFS and KWS in MKFRNP and yet these prevention measures would help to decrease the risk of catastrophic fires [7].

MKFRNP requires the best management strategy that will help communities, natural resource managers and other stakeholders to address both damaging and beneficial fires within the context of the natural environments and socio-economic systems in which they occur, by evaluating and balancing the relative risks posed by fires with the beneficial ecological and economic effects they may cause in a given conservation area, landscape, or region [4]. It should help to identify factors influencing fire ignition as it relates human needs and land use activities to factors influencing fire ignition. The best management strategy should also estimate and ascertain the roles of external drivers in influencing fire danger and the positive and negative effects of fires [1,3,4]. The best management strategy should also help in evaluating the benefits and risks of different management activities and developing fire management guidelines considering human needs and land use activities [4].

*2.2. Methodological Approach of the Study*

In this study, the methodological approach was classified into seven steps as shown in Figure 2. The facilitators comprising scientific experts from Egerton University, University of Natural Resources and Life Sciences, Vienna (BOKU) and Kenya Forest Research Institute (KEFRI), the KFS managers, KWS managers, the chief Ecosystem conservator, CFAs, NGOs, county water regional officers, other stakeholder groups and the focus group discussions supported the design of the management strategies and the set of O&C for the evaluation. The AHP was applied by a team of experts to select the best management strategy that helps to reduce fire danger and increase the benefits obtained in MKFRNP to fulfill multi-stakeholder interests. The individual steps are described as follows:

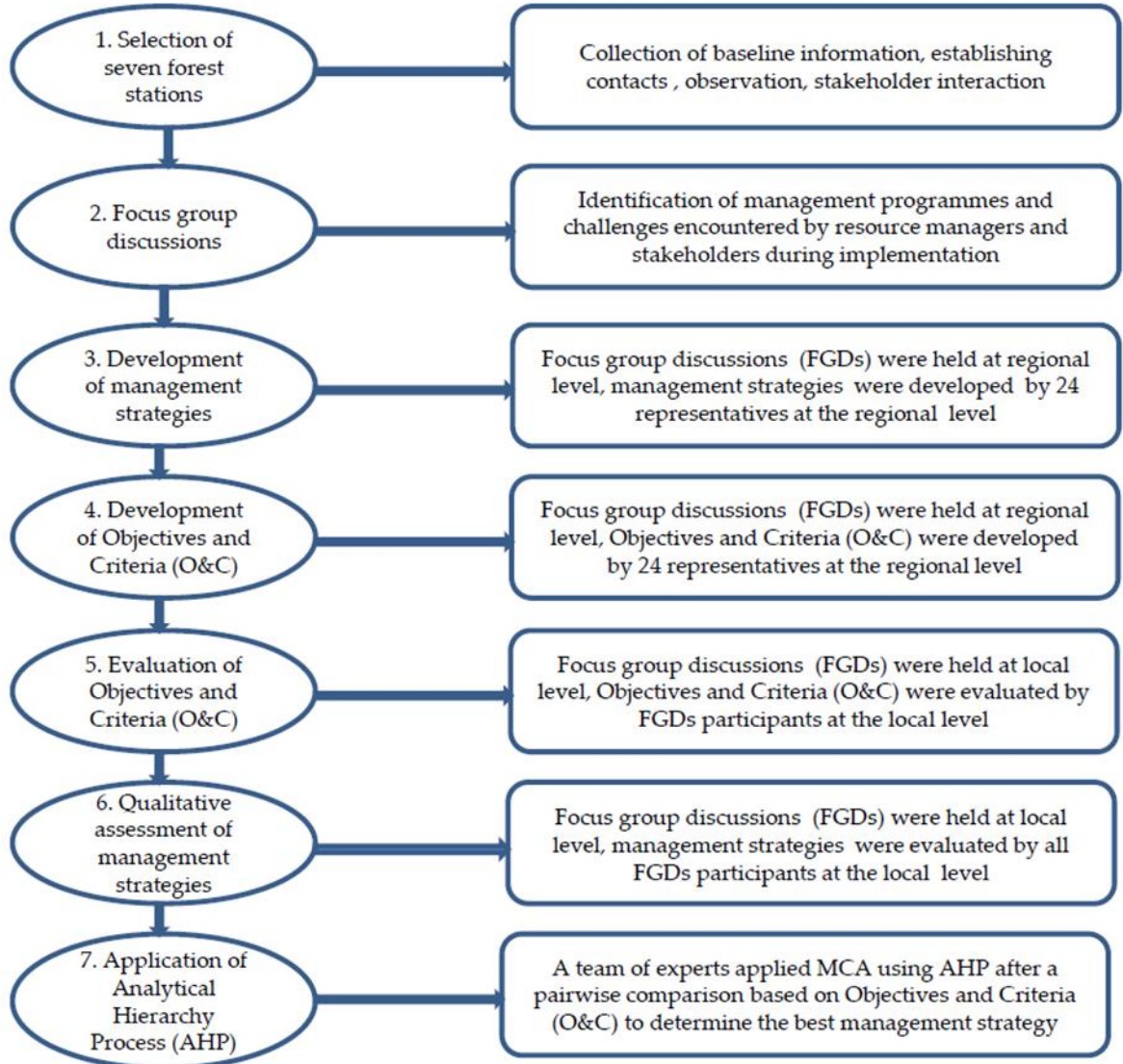

**Figure 2.** The methodological approach for selecting the best management strategy in Mount Kenya Forest Reserve and National Park (MKFRNP).

2.2.1. Selection of the Seven Forest Stations

In the first stage, the MKFRNP environment was described by collecting background information on management policies, socio-economic activities, and the bio-physical assessments were also done. Then contacts with different key stakeholders were established. This helped to understand the existing rules and regulations that govern decision making processes about resource allocations, such as, provisions that enhance the livelihoods of the local communities and opportunities for income generation activities and existing fire threats to the MKFRNP. The MKFRNP is divided into 23 forest stations and it was not possible to visit all of them during the study because it would have been too costly and time-consuming. Seven forest stations in MKFRNP were then selected based on the description of the current management objectives, benefits obtained by stakeholders and the number of fire incidences recorded by KFS and KWS from 1980 to 2015. The selected forest stations with a very high fire danger are Marania, Ontulili and Gathiuru having experienced 49, 71 and 63 fires incidences respectively in the last 35 years. Nanyuki and Naru Moru have a high to moderate fire danger having experienced 18 and 5 fire incidences respectively. Hombe and Chehe are considered as low fire danger forest stations, each having experienced only one fire incidence in the same period of time. Basic

information was gathered on the current management programmes that are being implemented by the KFS and KWS in the seven selected forest stations in MKFRNP as shown in Table A1 in Appendix A.

### 2.2.2. Focus Group Discussions (FGDs)

A focus group discussion is a productive and positive way to gather people together with similar backgrounds or experiences to discuss a specific topic of interest [19]. There was a need to hold FGDs with various stakeholders living around, working in or with interests in MKFRNP to address the issues affecting the current management. FGDs were then held at the regional level and local levels in all seven forest stations selected for this study, and facilitators from Egerton University, University of Natural Resources and Life Sciences, Vienna (BOKU) and Kenya Forest Research Institute (KEFRI) supported the process. At the regional level, the Chief Ecosystem Conservator, seven KFS forest managers, seven KWS wildlife managers, seven rangers, seven university specialists and one staff member of KEFRI were selected to attend the FGDs based on their gender, level of knowledge and expertise on issues affecting community development, agriculture, forestry, water and wildlife management [5]. The regional FGDs helped to identify the existing management strategies in MKFRNP. At the local level, 21 CFA members were selected in each of the seven forest stations to attend the FGDs. This was mainly based on gender, education, leadership roles in CFAs, experience in firefighting, the needs and benefits that they obtained in MKFRNP. Other community members, village leaders, NGOs, religious groups youth representatives and women representatives were also invited to attend the FGDs to express the opinions of other community user groups that obtain benefits in MKFRNP [20]. The local level FGDs participants with the help of facilitators identified and discussed the challenges affecting the implementation of the existing management strategies and expressed their preferences based on their needs as shown in Table A2 in Appendix B.

### 2.2.3. Development of Management Strategies (MS)

In the third stage, FGDs were held at the regional level by the seven representatives and facilitators to discuss the existing management programmes and the challenges resource managers were facing in making management decisions on how to reduce fire danger and increase the benefits obtained by the different stakeholders in MKFRNP. Resource managers and stakeholders are facing difficulties to ensure that rare and threatened species, as well as their habitats, are protected. There is a strong need to preserve and restore the existing habitats and improve ecosystem connectivity between them [5,15]. On the other hand, wood and non-wood natural forest products should be sustainably exploited and the commercial production of timber and other forest products should be favored. Plantation forests might help to meet the market demands and restore degraded forest areas. Local communities should be actively involved in forest management activities [3]. MKFRNP plays a critical role in water catchment functions in Kenya and there is a strong need to restore and conserve the water catchment area and control the water collection from rivers [1,3,21]. The MKFRNP offers diverse low impact tourist activities, which help to augment resource protection. Therefore, tourist products and services can be marketed and adequate visitor accommodation facilities can be developed [22,23]. The FGDs participants also stated that during the implementation of the Community Partnership and Education Programme, the resource managers were facing difficulties in improving communication among the people living in the area and minimizing human-wildlife conflicts in adjacent areas [5,20,24]. In this context, staff welfare and motivation are critical components for the success of the conservation efforts and that effective and efficient management infrastructure needs to be provided [1,3,7]. Additionally, it was stated that security presence should be extended across MKFRNP and collaboration with key stakeholders in security matters should be strengthened to ensure that encroachment, marijuana cultivation, accidental forest fires, poaching of wild animals, illegal logging, and other forms of illegal activities are minimized [4,7,23]. After discussing the challenges in natural resource management, the strategies were outlined, describing how to overcome the main problems, ensure stakeholder

participation, as well as minimize the perceived future threats. The regional FGDs described and developed nine management strategies with the help of the facilitators in detail for further evaluation.

### 2.2.4. Developing Objectives and Criteria

In the fourth step, a set of objectives and criteria (O&C) were identified at the regional FGDs over three days. They helped to generate an initial set of O&C for evaluating and selecting the best management strategy that will help to reduce fire danger and increase the benefits obtained by different stakeholders in MKFRNP. A top-down approach was used to ensure that information gathered in the local FGDs was not lost [12]. The O&C set was developed based on the information and experiences affecting community development, agriculture, forestry, water and wildlife management in MKFRNP. Based on the inputs from the seven representatives and stakeholders as well as different international and national examples relevant objectives were identified. Then criteria were defined to decompose the objectives. Finally, the regional FGDs selected 12 objectives and 28 criteria for further discussion and evaluation with participants at the local level in the forest stations. The bottom-up approach was organized in a way to accommodate direct involvement and participation of various stakeholders with interests in the MKFRNP so as to secure their commitment in the long term. In all local level FGDs the participants adapted the O&C to their local conditions and criteria that were less preferred or redundant were excluded from the set. At the end, a final set of 8 objectives and 21 criteria was proposed for the evaluation of the management strategies (Figure 3).

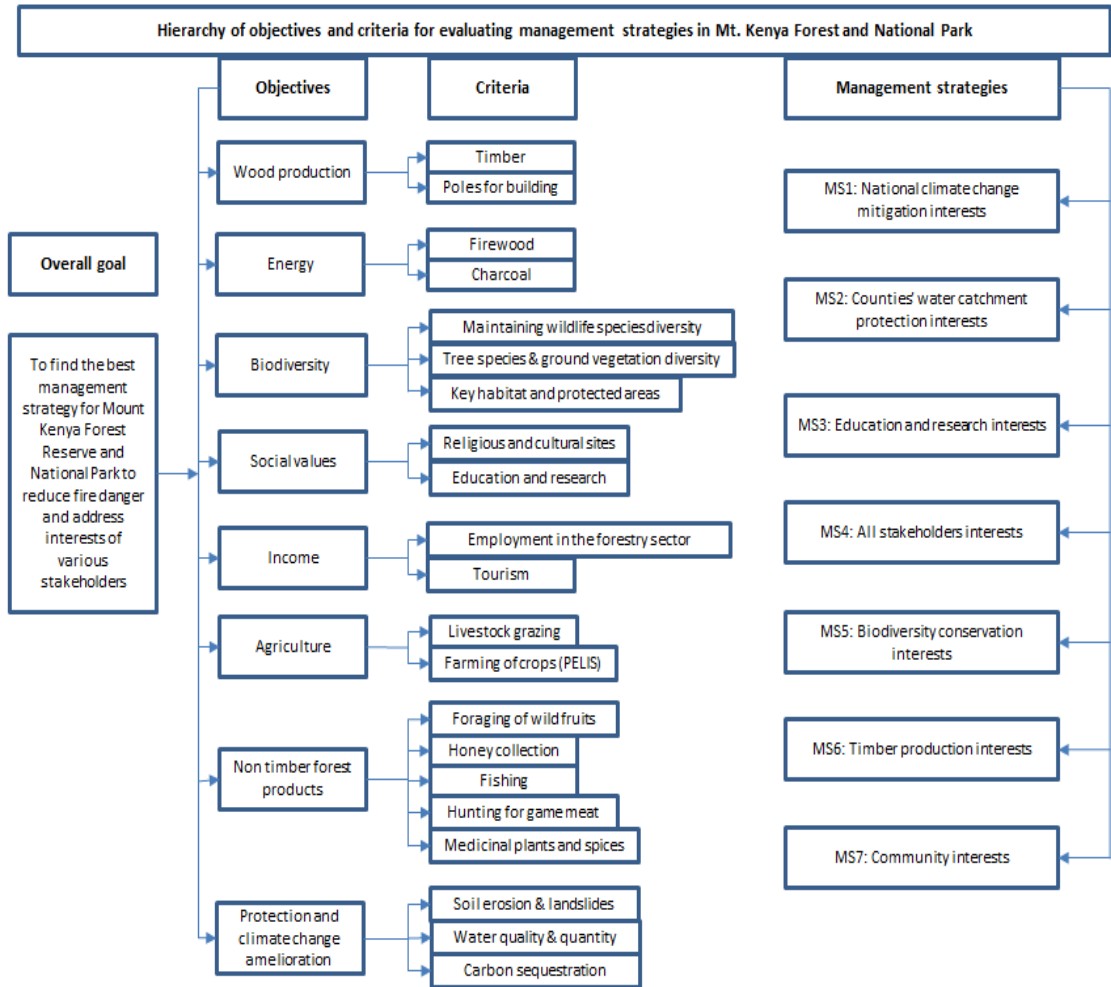

**Figure 3.** Evaluation hierarchy of the objectives and criteria for selecting the best management strategy in MKFRNP.

### 2.2.5. Elicitation of Preferences

In the fifth step, the bottom-up approach was used to identify the preferences of the regional and local FGDs participants from the seven forest stations in MKFRNP. A brief introduction about the need for a participatory evaluation of the O&C by all FGDs participants was given by the facilitators. The participants were divided according to their level of expertise and interests, so as to obtain priorities and minimize cases of individuals or certain groups trying to dominate the discussions during local FGDs [25]. Some FGDs' participants were very confident in expressing the importance of certain O&C, while others were challenged by the need to select the most relevant objectives. At the end of the evaluation process, the FGD's participants in each forest station had expressed their priorities by scoring (from 0 to 100) the objectives and criteria at each level. The scores provided by the FGDs participants were later on used to calculate the mean values for each criterion in the seven forest stations in MKFRNP.

### 2.2.6. Qualitative Assessment of the Developed Management Strategies

At the sixth stage of the local FGDs, the communities were supported by the facilitators to understand how the human activities to obtain several goods and services in MKFRNP are causing wildfires, degradation, influencing tree species composition and endangering wildlife populations. The participants discussed the strengths and weaknesses of the nine developed management strategies. They all shared different views as to which management strategy would best fulfill the different interests of all the stakeholders, overcome the main problems, ensure participation in decision making, as well as minimize the perceived future threats. After intensive discussions, the regional and local FGDs finally selected seven management strategies with the help of the facilitators (Table A1 in Appendix A). Based on their input, the seven management strategies were assessed qualitatively with the help of experts. They evaluated the effect of each management strategy on how it would reduce the fire danger in MKFRNP according to the following statements: "to a very great extent", "to a great extent", "to some extent", "to a little extent", "to a very little extent" or "to no extent" (Table 1). Additionally, the effect of the management strategies on the benefits obtained by stakeholders in MKFRNP was qualitatively assessed based on the defined set of objectives and criteria according to the following statements: "has a positive impact", "has a negative" or "has no impact" (Table 2). The qualitative assessment of the seven management strategies provided the input for the evaluation with the AHP.

**Table 1.** Evaluation of the seven management strategies and their Integrated Fire Management (IFM) activities qualitatively with regard to objectives and criteria.

| IFM Activities Management Activities | Targets | MS1 | MS2 | MS3 | MS4 | MS5 | MS6 | MS7 |
|---|---|---|---|---|---|---|---|---|
| 1. Increase stakeholder participation in IFM decision making; | 1.1-Government departments and ministries, | +++++ | ++++ | ++++ | +++++ | +++ | +++ | ++ |
| | 1.2-Communities, | ++ | + | + | +++++ | +++++ | + | +++++ |
| | 1.3-International agencies, | +++++ | +++ | +++ | +++++ | ++++ | + | +++ |
| | 1.4-NGOs, | +++ | + | + | +++++ | +++++ | + | +++++ |
| | 1.5-Conservationists | +++ | + | +++ | +++++ | +++++ | + | +++++ |
| 2. Reduce fire hazards and danger (particularly in and around communities and other high value areas) | 2.1-Clean up dry litter accumulations | +++ | ++++ | 0 | ++++ | +++++ | +++ | + |
| | 2.2-Close fire prone areas in dry season | +++++ | +++++ | 0 | +++++ | +++ | ++++ | + |
| | 2.3-Handle inflammable materials safely | +++++ | +++++ | 0 | +++++ | +++ | ++++ | + |
| | 2.4-Establish firebreaks and forest roads | +++++ | +++++ | 0 | +++++ | +++ | +++++ | + |
| | 2.5-Provide adequate equipment | +++++ | +++++ | 0 | +++++ | +++ | +++ | + |
| | 2.6-Train fire crews | +++++ | +++++ | +++++ | +++++ | +++ | ++++ | + |
| | 2.7-Establish less fire prone vegetation | +++++ | + | + | +++ | +++ | ++ | + |

**Table 1.** *Cont.*

| IFM Activities Management Activities | Targets | MS1 | MS2 | MS3 | MS4 | MS5 | MS6 | MS7 |
|---|---|---|---|---|---|---|---|---|
| 3. Carefully use prescribed burning where the benefits are clearly defined; | 3.1-Establish fire lines | +++++ | ++++ | +++ | +++++ | +++ | +++ | + |
| | 3.2-Monitor fuel and weather conditions | +++++ | ++++ | ++++ | +++++ | ++++ | +++ | + |
| | 3.3-Controlled burning of agricultural lands | + | + | + | ++++ | ++++ | + | +++++ |
| | 3.4-Controlled burning of grassing grounds | + | + | + | ++++ | +++++ | + | +++++ |
| | 3.5-Controlled burning of timber slash | + | + | + | ++++ | + | + | + |
| 4. Monitor & manage fire on communities land and forests; | 4.1-Construct look out towers | +++++ | +++ | 0 | +++++ | +++ | +++ | + |
| | 4.2-Deploy fire monitoring crew/ scouts | +++++ | ++++ | 0 | +++++ | ++++ | +++ | +++ |
| | 4.3-Establish access to water sources | +++++ | +++++ | 0 | +++++ | +++ | +++ | + |
| | 4.4-Evacuate people | +++++ | +++++ | 0 | +++++ | + | +++ | + |
| 5. Integrate fire management programs that control invasive plant species; | 5.1-Prevention | +++++ | ++++ | + | ++++ | +++++ | +++ | + |
| | 5.2-Chemical control | ++ | ++ | + | ++ | ++++ | + | + |
| | 5.3-Manual control | +++++ | ++++ | + | +++++ | +++++ | +++ | +++ |
| | 5.4-Cultural control/ competition | +++ | +++ | + | ++++ | +++++ | +++ | + |
| | 5.5-Biological control | +++ | +++ | + | +++ | +++++ | + | + |
| 6. Minimize outbreaks of non-ecological fires in hydrophobic soils; | 6.1-Protection plans | +++ | +++ | + | +++++ | +++++ | +++ | + |
| | 6.2-Protection maps | ++++ | ++++ | + | +++++ | +++++ | +++ | + |
| | 6.3-Prevention of erosion | ++++ | +++++ | 0 | +++++ | ++++ | +++ | ++ |
| | 6.4-Prevention of loss of organic-rich soils | +++++ | ++++ | 0 | ++++ | ++++ | +++ | ++ |
| 7. Incorporate land use & forest managers, CFAs and policy actors in IFM | 7.1-Land use planning | +++++ | +++ | + | +++++ | ++++ | +++ | ++ |
| | 7.2-Forest resource management planning | ++++ | + | ++ | +++++ | ++++ | ++++ | + |
| | 7.3-Community participation in IFM | + | + | + | +++++ | +++ | + | +++++ |
| | 7.4-Laws, policy, institutional framework | +++++ | +++++ | ++ | +++++ | ++++ | ++++ | + |
| 8. Develop a high level of public awareness and support for IFM; | 8.1-Public meetings and social groups | +++++ | ++++ | + | +++++ | ++++ | +++ | +++++ |
| | 8.2-Posters & sign boards | +++++ | +++++ | ++ | +++++ | ++++ | +++ | +++ |
| | 8.3-Radio | +++++ | ++++ | + | +++++ | ++++ | +++ | +++ |
| | 8.4-TV | +++++ | ++++ | ++ | +++++ | ++++ | +++ | +++ |
| | 8.5-Newspapers | +++++ | ++++ | +++ | +++++ | +++ | +++ | + |
| | 8.6-Internet | ++++ | +++ | +++++ | +++++ | +++ | +++ | ++ |
| 9. Incorporate traditional fire use and management practices when developing and implementing of IFM strategies; | 9.1-Clearing land for PELIS | +++ | 0 | 0 | +++++ | 0 | ++++ | +++++ |
| | 9.2-Replenishing nutrients on farms | + | 0 | + | +++++ | 0 | + | +++++ |
| | 9.3-Killing woody species in rangelands | + | + | 0 | +++++ | + | + | +++++ |
| | 9.4-Encouraging grass growth | + | + | 0 | +++ | ++++ | + | +++++ |
| | 9.5-Increasing wild seed production | + | + | 0 | +++ | ++++ | + | +++++ |
| | 9.6-Honey collection | + | +++ | 0 | ++++ | ++ | + | +++++ |
| | 9.7-Hunting | + | +++ | 0 | +++ | +++++ | + | +++++ |
| 10. Reducing IFM costs | 10.1-Staff salaries | +++ | +++ | + | ++ | +++++ | +++ | ++++ |
| | 10.2-Equipment purchase | +++ | + | 0 | +++ | +++++ | ++ | ++ |
| | 10.3-Repair and maintenance | +++ | + | 0 | +++ | +++++ | +++ | ++ |
| | 10.4-Fuel costs | +++ | + | 0 | +++ | +++++ | ++ | +++ |

+++++ to a very great extent, ++++ to a great extent, +++ to some extent, ++ to a little extent, + to a very little extent, 0 to no extent.

According to the evaluation of the seven management strategies (Table 1) and their contribution to Integrated Fire Management activities MS4 (all stakeholder interests) showed the highest number of activities followed by MS5 (biodiversity conservation interests) and MS1 (national climate change mitigation interests). For the qualitative assessment of the management strategies according to the

objectives and criteria to increasing the benefits obtained in MKFRNP the stakeholders indicated that MS3 (Education and research interests) has almost no impact, while MS6 (timber production interests) showed even several negative impacts (Table 2).

**Table 2.** Qualitative assessment of the seven management strategies (MS) based on objectives and criteria (O&C) to determine how they contribute to increasing the benefits obtained in MKFRNP.

| Objectives | Criteria | MS1 | MS2 | MS3 | MS4 | MS5 | MS6 | MS7 |
|---|---|---|---|---|---|---|---|---|
| 1. Wood production | 1.1 Timber | + | - | 0 | + | - | + | + |
| | 1.2 Poles | + | - | 0 | + | - | + | + |
| 2. Energy | 2.1 Firewood | - | - | 0 | + | - | + | + |
| | 2.2 Charcoal | - | - | 0 | - | - | - | + |
| 3. Biodiversity | 3.1 Maintaining wildlife species diversity | 0 | + | + | + | + | - | - |
| | 3.2 Maintaining vegetation species diversity | + | + | + | + | + | - | - |
| | 3.3 Key habitat and protected areas | + | + | 0 | + | + | - | - |
| 4. Social values | 4.1 Religious and cultural sites | 0 | 0 | 0 | + | + | 0 | + |
| | 4.2 Education and research | + | + | + | + | + | 0 | 0 |
| 5. Income | 5.1 Employment in the forest sector | + | + | + | + | - | + | + |
| | 5.2 Tourism | 0 | 0 | 0 | + | + | - | + |
| 6. Agriculture | 6.1 Farming of crops (PELIS) | + | - | 0 | + | 0 | + | + |
| | 6.2 Livestock grazing | - | - | 0 | + | - | + | + |
| 7. Non timber forest products | 7.1 Foraging wild fruits | 0 | 0 | 0 | + | + | - | + |
| | 7.2 Honey collection | + | + | 0 | + | + | - | + |
| | 7.3 Fishing | + | + | 0 | + | + | - | + |
| | 7.4 Hunting of game meat | - | - | 0 | - | - | - | + |
| | 7.5 Medicinal plants and spices | + | + | + | + | - | - | + |
| 8. Protection and climate change amelioration | 8.1 Soil erosion and landslides | + | + | 0 | + | + | + | + |
| | 8.2 Water quality and quantity | + | + | 0 | + | + | - | + |
| | 8.3 Carbon sequestration | + | + | 0 | + | + | + | - |

+ has a positive impact, - has a negative impact, 0 has no impact.

### 2.2.7. Final Evaluation of the Management Strategies with the Analytical Hierarchy Process (AHP)

With the AHP, pairwise comparisons are made among a defined set of alternative options with regard to an evaluation hierarchy, to provide a cardinal ranking of the alternatives [14]. The technique allows the consistency of the decision makers' evaluations to be checked, thus reducing a potential bias in the elicitation process. Since some of the criteria are always contrasting, the best option is not the one which optimizes each single criterion, rather the one which achieves the most suitable trade-off among the different criteria [14]. The AHP was used to calculate weights for all objectives and criteria of the defined evaluation hierarchy (O&C) based on the preferences that were expressed by the FGDs participants. The higher the weight, the more important the corresponding criterion will be. Finally, the AHP allowed combining the preferences of the seven management strategies with the criteria weights and thus determining a global priority for each strategy.

The pairwise comparisons were not carried out at the field level, due to their complexity, as other studies have described problems related to their time-consuming nature [12]. The mean values for each criterion in the seven forest stations were calculated by the experts based on the scores provided by the participants of the FGDs. The mean values of the scores derived from the FGDs' participants had to be transferred to pairwise comparisons. The scores provided for the objectives and criteria were used to calculate the preferences of the objectives and criteria in the AHP, assuming that the objectives and criteria with a highest score are of higher importance. The qualitative assessment, which provided for the evaluation of the performance of the management strategies (Table 2) was used to derive the preferences for the seven options. The options were ordered according to the qualitative assessment and the preference values were calculated for each strategy for each criterion in the hierarchy using the Expert Choice Software. This helped to identify the best performing management strategies and potential trade-offs with regard to different preferences.

## 3. Results

### 3.1. Scoring of Objectives and Criteria by FGDs Participants

The scores (0–55) represent the mean values for each criterion in the seven forest stations that were calculated by the experts based on the scores provided by the participants in the FGDs that were held at the local level. It was found out that the objectives with the highest mean values of the scores are of higher importance. The scores indicate that in all forest stations, firewood got the highest scores, followed by farming (PELIS) and livestock grazing. Water quality/quantity and maintaining vegetation species diversity are ranked third place, timber harvesting was ranked fourth, employment in the forest sector was ranked fifth and tourism was ranked sixth. However, fishing was ranked second last and hunting was ranked last (Table 3).

**Table 3.** Importance of the Objectives and Criteria in the seven forest stations based on the scores provided by the participants in the focus group discussions (FGDs).

| Current Management Objectives | Benefits Obtained by Various Stakeholders in MKFRNP | Forest Stations According to Their Fire Danger | | | | | | |
|---|---|---|---|---|---|---|---|---|
| | | Very High Fire Danger | | | High Fire Danger | Moderate Fire Danger | Low Fire Danger | |
| | | Marania | Ontulili | Gathiuru | Nanyuki | Naru moru | Hombe | Chehe |
| 1. Wood production | 1.1 Timber | 40 | 25 | 30 | 25 | 35 | 30 | 0 |
| | 1.2 Poles for building | 15 | 20 | 20 | 20 | 15 | 10 | 5 |
| 2. Energy | 2.2 Firewood | 45 | 45 | 45 | 45 | 50 | 45 | 55 |
| | 2.3 Charcoal | 5 | 5 | 5 | 5 | 10 | 5 | 5 |
| 3. Biodiversity conservation | 3.3 Maintaining wildlife species diversity | 25 | 25 | 25 | 25 | 25 | 25 | 30 |
| | 3.4 Maintaining vegetation species diversity | 25 | 35 | 25 | 35 | 35 | 30 | 35 |
| | 3.5 Key habitat & protected areas | 20 | 15 | 20 | 20 | 15 | 15 | 30 |
| 4. Social values | 4.1 Religious & cultural sites | 5 | 5 | 5 | 10 | 10 | 5 | 25 |
| | 4.2 Education & research | 5 | 5 | 5 | 5 | 5 | 5 | 5 |
| 5. Income | 5.1 Employment in forestry sector | 20 | 30 | 35 | 20 | 25 | 30 | 35 |
| | 5.2 Tourism | 30 | 20 | 20 | 30 | 20 | 20 | 5 |
| 6. Agriculture | 6.1 Livestock grazing | 35 | 30 | 35 | 35 | 40 | 30 | 35 |
| | 6.2 Farming of crops (PELIS) | 35 | 40 | 45 | 35 | 30 | 45 | 0 |
| 7. Non timber forest products | 7.1 Foraging of wild fruits | 5 | 10 | 5 | 5 | 5 | 10 | 10 |
| | 7.2 Honey collection | 20 | 20 | 15 | 25 | 5 | 15 | 20 |
| | 7.3 Fishing | 5 | 0 | 5 | 5 | 5 | 0 | 0 |
| | 7.4 Hunting of game meat | 0 | 0 | 0 | 5 | 0 | 0 | 0 |
| | 7.5 Medicinal plants & spices | 5 | 10 | 5 | 5 | 5 | 10 | 10 |
| 8. Protection, water and climate change amelioration | 8.1 Prevention of soil erosion & landslides | 25 | 15 | 15 | 20 | 20 | 20 | 35 |
| | 8.2 Water quality and quantity | 30 | 30 | 35 | 25 | 35 | 25 | 35 |
| | 8.3 Carbon sequestration | 15 | 15 | 15 | 10 | 10 | 15 | 10 |

### 3.2. Preferences of Objectives and Criteria for Different Fire Danger

The priorities of the objectives according to the fire danger categories (reference scenario, very high fire risk, high fire risk, moderate fire risk, and low fire risk) vary. For the "reference" scenarios of all the eight objectives were equally weighted (1/8 = 0.125). We found that the biodiversity conservation objective was ranked first for all four scenarios, agriculture was ranked second for forest stations with a very high and moderate fire danger, climate change amelioration was ranked second for forest stations with a high and a low fire danger (Figure 4).

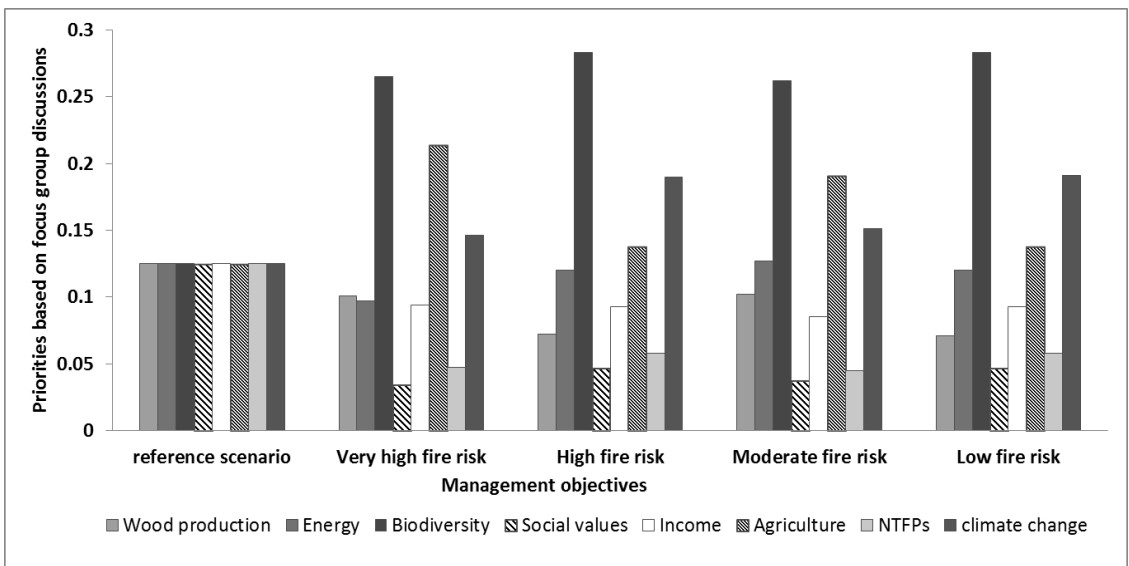

**Figure 4.** Priorities of management objectives according to the fire danger categories.

The priorities of the criteria according to the fire danger categories (reference scenario, very high fire risk, high fire risk, moderate fire risk, and low fire risk) also vary. We found that firewood, timber, livestock grazing and employment in the forest sector are highly preferred criteria in the seven forest stations according to the preferences derived by the PWC with the AHP (Table A3 in Appendix C).

### 3.3. Priorities of the Management Strategies

The priorities of the management strategies according to the fire danger categories (reference scenario, very high fire risk, high fire risk, moderate fire risk and low fire risk) are presented in Figure 5. We found out that although the seven forest stations had a different fire danger, all the FGDs participants rated MS7 (community interests) first based on the weights given to all objectives and criteria. Only in the reference scenario (with equal weights for all objectives) was the MS5 (biodiversity conservation) ranked first. The second and third ranked strategy for the four different fire danger categories were MS5 (biodiversity conservation) and MS4 (all stakeholders interests) respectively. The MS3 (education and research) was classified as the least preferred strategy in each case.

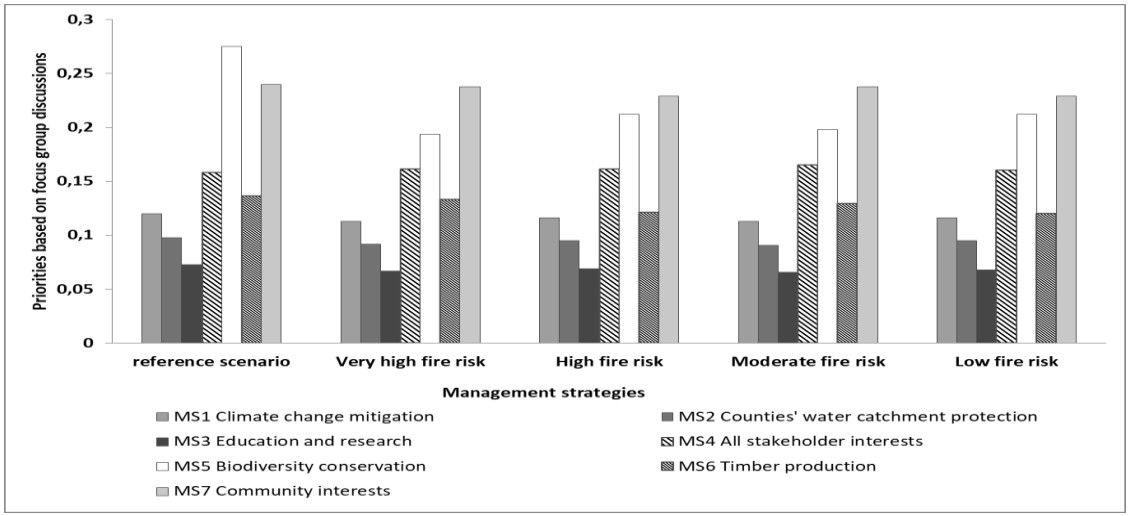

**Figure 5.** Priorities of the seven management strategies for the reference scenario and the four fire danger categories.

## 4. Discussion

### 4.1. Use of MCA in the Evaluation of Management Strategies

Our analysis provides important insights into the application of a MCA approach to develop management strategies, objectives and criteria that can be used to identify, structure, monitor and evaluate the best management strategy that will help to reduce fire danger and increase the benefits obtained in MKFRNP. Other studies have also shown that MCA is both an appropriate and useful approach for capturing diverse views, objectives and perspectives of different stakeholders involved in decision making [14].

The careful selection of FGD participants during the study, helped to develop and evaluate the seven management strategies based on the identified objectives. A team of professional experts in the application of MCA techniques was selected and made responsible for supporting the process of comparing management strategies. The interaction between the facilitators, FGDs' participants, and experts was undertaken by having a high number of meetings within a short period of time [26]. However, it was found out that the FGDs' participants were had difficulties in expressing their preferences with regard to the management strategies, and importance of objectives and criteria. Many of them were not familiar with the forestry and wildlife terms that were used in the qualitative evaluation, while others did not understand well how the scoring and ranking of the developed management strategies, objectives and criteria was to be done and this made the exercise challenging and time-consuming. The developed management strategies varied both in the temporal and spatial scale to meet the various stakeholder interests and this made it even more difficult for the participants to evaluate the strategies. Another observation was that some FGDs' participants only preferred objectives that required a shorter time to be achieved and therefore had strong interests in having them implemented [9,27]. This indicates that the best management strategy that is applicable on the entire area of MKFRNP has to help reduce the fire danger and consider both the short term and long-term interests of the different stakeholder groups for it to be accepted and implemented [9,13].

The facilitators used a mixture of a bottom-up and a top-down approaches during the regional and local FGDs. This allowed for keeping a consistent overall framework for the evaluation, while including inputs from the participants in a participatory way. The FGDs' participants were able to express their own preferred management strategies, objectives and criteria and appropriately address some of the challenges in the qualitative assessment. This helped to accurately structure the problem, increase transparency and improve the quality of the decision-making process by contributing to a participatory implementation [9,12].

The use of MCA for evaluating the management strategies for implementation in MKFRNP helped to address the stakeholder interests and provided a framework for evaluating trade-offs in a transparent and understandable way [9,28,29]. The application of the MCA allowed to come up with solutions which resulted in higher level of overall stakeholder satisfaction [12].

Several studies have shown that the use of the AHP can impose several challenges, as it can be time-consuming when scores and ranking of the participants are transferred to pairwise comparisons [9,25,30]. AHP allows the use of both qualitative and quantitative information when comparing the performance of alternatives [14]. However, very often it is not possible to consider quantitative information in assessing management strategies with regard to the effectiveness to fire management or in improving the livelihood conditions. Therefore, transferring the qualitative ratings to pairwise comparisons is useful [12]. The analysis with the AHP allowed for the sensitivity of each management strategy to be identified by varying the weights assigned to each objective.

### 4.2. Performance of Management Strategies

MS1 (national climate change mitigation interests), MS5 (biodiversity conservation interests) and MS6 (timber production interests) are long term management strategies. MS2 (counties' water catchment protection interests), MS3 (education and research interests), MS4 (all stakeholder interests)

and MS7 (community interests) are short-term management strategies. However, the results indicate that MS5 and MS7 are almost equally preferred. This indicates that long and short-term aspects are considered as relevant by the various stakeholders. To achieve sustainable management of MKFRNP, other programmes should offer a similar degree of importance for the improvement of community livelihoods [31]. Objectives such as wood production, biodiversity conservation, protection, and climate change amelioration require long-term management to provide positive outcomes and meet various stakeholder interests. However, energy, social values, income, agriculture and non-timber forest products require short-term management activities [9,32].

The current management strategies being implemented by resource managers have focused more on the conservation of biodiversity and have paid less attention to fulfilling other objectives such as wood production, energy, social values, non-timber forest products, protection and climate change amelioration [33]. The KFS and CFAs have been implementing the Mount Kenya Forest Reserve (MKFR) management plan 2010–2019 that considers sustainable management, including conservation and rational utilization of the forest resources for socio-economic development [1]. On the other hand, the KWS has been implementing the Mount Kenya Ecosystem (MKE) management plan 2010–2020 that has Ecological Management programme that aims at addressing biodiversity restoration and protection, linking ecosystems, and carrying out applied research to understand how the ecosystem functions [1]. Biodiversity conservation is seen as basis for the functionality of MKFRNP ecosystem including fuel wood, soil fertility, water, timber, poles, wildlife, tree species, agriculture, non-timber forest products, protection, climate change amelioration, culture and scenery [9,34,35]. Since MS5 is effective in conserving biodiversity, reducing fire danger and increasing the benefits, it has the greatest likelihood of being socially, economically and politically acceptable. When MS5 is fully implemented in MKFRNP, it is possible that many of the current management problems might be reduced [9]. However, although the biodiversity conservation objective is ranked first by all the participants of the FGDs, the results of the analysis indicate that MS7 (community interests) is the most preferred management strategy followed by the MS5 (biodiversity conservation interests). This means that the resource managers will have to work closely with all stakeholders so that the selected management strategy addresses threats like poaching of wildlife and control of wildfires, which threatens all conservation targets and requires long-term monitoring as well [1].

Agriculture was ranked as the second most relevant objective and therefore it is obvious that MS7 (community interests) was more preferred by the FGDs' participants. This is in line with other studies that have shown that communities in developing countries are more concerned with socio-economic activities aimed at achieving their livelihood benefits such as employment, farming activities, firewood collection, water collection, grazing, honey collection, tourism, herbal medicine, hunting or timber production [36–38]. During preference elicitation, the FGDs participants scored hunting low because it is prohibited by law to hunt in MKFRNP. Some community members have been prosecuted for being involved in wildlife poaching, by getting imprisoned and or paying heavy fines [5]. Most of the community members living around MKFRNP prefer growing food crop, cash crop and keeping livestock and these are reasons why fishing was scored low by FGDs participants [4]. Fishing in MKFRNP's rivers, dams and lakes is legal but only a few small-scale farmers around MKFRNP have initiated fish farming projects for the growing market especially in the local hotel industry [4]. The growing population and the increasing human demands pose a serious question as to how long this management strategy may address the growing stakeholder interests [5]. The findings of this study show that the decision-makers and policy actors need to consider biodiversity and community interests in the decision-making process. This will allow them to select and implement the best management strategy that reduces fire danger and increases the benefits obtained in MKFRNP.

## 5. Conclusions

This is the first time that O&C have been developed for assessing management strategies in MKFRNP through a participatory process with all stakeholders. Due to the shortage of time, limited

number of experts in stakeholder groups and the limited number of reliable data sources available, this limits the results of our study to some extent. However, because of the quite robust results, it can be assumed that the findings are applicable to other forests stations as well. It is possible to adapt the evaluation framework and revise the management strategies based on a more widely-based discussion with different stakeholder groups. Applying O&C assessments for the sustainable management of MKFRNP has the potential to reduce the information gap between decision-makers at the local and national levels.

In the year 2014, the parliament of Kenya passed the county governments' fire and disaster management bill that prepared the ground for the country to establish and implement integrated fire management approaches in the future [4]. Even though the Kenya Grass Fire Act, Cap 327, provides a regulation for planned burnings of bushes, shrubs, grass, crops, and stubble within protected areas, the KFS and KWS have continued to practice fire suppression campaigns instead of using prescribed burning activities to manage fuel accumulation in MKFRNP [4]. This is mainly based on the belief that any disturbance, such as fire, disrupts the progress towards an equilibrium state [4]. Total fire suppression and other human-caused environmental changes have resulted in huge and catastrophic wildfires in MKFRNP [7].

The performance of the management strategies might be different in other forests and national parks in Africa, where conditions are slightly different and where different views of stakeholders may be present. However, our study presents recommendations for further policy options that consider forest health, productivity and socio-economic values, as basic requirements for improving the livelihoods of the people. Moreover, forest and wildlife management need to take into account how the involvement of the local communities in the decision-making process could be developed, with the main goal of stimulating the development of commonly accepted management strategies for MKFRNP. Resource managers can make better management decisions in the future to ensure that: rare and threatened species are protected, restored and monitored; habitats are protected, preserved and restored; ecosystem connectivity is established to increase resilience; and Mt. Kenya ecosystem functioning is understood [5,15]. Further research needs to be carried out in other forest and national parks in Kenya, as different stakeholder interests, vegetation and wildlife species, and threats require adapted management strategies and a revised evaluation framework.

**Author Contributions:** K.W.N. and H.V. worked jointly on the study design, including guidelines for the focus group discussions; K.W.N. performed the interviews and facilitated the focus group discussions; K.W.N. and H.V. analyzed the data; K.W.N. wrote the paper; and H.V. contributed to the paper.

**Funding:** This research was funded by the Austrian Partnership Programme in Higher Education and Research for Development (APPEAR), Commission for Development Research (KEF) Austria grant number KEF P211 and the APC was funded by the OA publishing fund at BOKU library services.

**Acknowledgments:** We acknowledge the funds of the Commission for Development Research (KEF P211) and the APPEAR scholarship programme for providing us with financial support for the research. We thank the management of Egerton University for providing us with staff, office space, internet, printing and library services during the research period. We also thank the Kenya Forest Service, Kenya Wildlife Service and Kenya Forest Research Institute for providing us with the permission to conduct the research at Mt. Kenya, and their support with staff and records during data collection. We also acknowledge the Community Forest Associations (CFAs) for actively participating in interviews and focus group discussions during data collection.

**Conflicts of Interest:** The authors declare no conflict of interest. The founding sponsors had no role in the design of the study; in the collection, analyses, or interpretation of data; in the writing of the manuscript, and in the decision to publish the results.

## Appendix A. Developed Management Strategies (MS) and the Main Activities that Need to Be Implemented in MKFRNP

**Table A1.** Developed management strategies (MS) and the main activities that need to be implemented to reduce fire danger and increase the benefits obtained in MKFRNP.

| Developed Management Strategies (MS) | Management Activities in MKFRNP | Integrated Fire Management (IFM) Activities in MKFRNP |
|---|---|---|
| 1. MS1<br>Climate change mitigation interests | 1.1-Increasing the capacity of carbon sinks through reforestation for timber and poles;<br>1.2-Reducing deforestation by arresting and prosecuting those involved in illegal logging and encroachment of settlements in to MKFRNP<br>1.3-Educating communities on adaptation and mitigation of climate change through the local media and village meetings | **1. Increase stakeholder participation in IFM decision-making;**<br>1.1-Government departments and ministries,<br>1.2-Communities,<br>1.3-International agencies,<br>1.4-NGOs,<br>1.5-Faith based organizations (FBOs),<br>1.6-Conservationists |
| 2. MS2<br>Counties' water catchment protection interests | 2.1-Increasing the quantity, improving the quality of water by planting trees in deforested areas, by arresting and prosecuting those involved in illegal logging, encroachment of settlements and illegal cultivation in water catchments;<br>2.2-Ensuring the existence and implementation of watershed management regulations by the counties' water ministries and watershed management groups in conjunction with CFAs;<br>2.3-Managing sloping lands properly by planting trees, bamboo and grasses to reduce soil erosion and landslides in MKFRNP | **2. Reduce fire hazards and danger (particularly in and around communities and other high-value areas);**<br>2.1-Clean up litter and rubbish accumulations<br>2.2-Reduce fuel loads (deadwood, grass)<br>2.3-Close hazardous areas to use during periods of extreme fire weather conditions<br>2.4-Handle inflammable materials safely<br>2.5-Establish firebreaks<br>2.6-Construct forest roads<br>2.7-Provide adequate equipment<br>2.8-Train fire crews<br>2.9-Establish less fire prone vegetation |
| 3. MS3<br>Education and research interests | 3.1-Improving skilled scientific research capacities by collaborating with institutions of education and research like KEFRI, local and international universities;<br>3.2-Providing well defined information in precautionary and protective measures to resource managers on weather conditions (droughts, temperature, precipitation, storms), pests, diseases, fires and invasive species;<br>3.3-Exchanging of technology and expertise knowledge on how to use modern forestry equipment, provide open access to education and research information on MKFRNP | **3. Carefully use prescribed burning where the benefits are clearly defined and the danger can be cost-effectively managed;**<br>3.1-Establish fire lines<br>3.2-Monitor weather conditions<br>3.3-Monitor fuel conditions<br>3.4-Controlled burning of agricultural lands<br>3.5-Controlled burning of grassing grounds<br>3.6-Controlled burning of timber slash |
| 4. MS4<br>All stakeholder interests | 4.1-Promoting ownership and user rights by providing equal opportunities to all stakeholder groups through registration and provision of licenses;<br>4.2-Strong law enforcement capacity by employing more rangers, fire patrol crews to arrest and prosecute all people involved in activities such as poaching of wildlife, illegal timber logging, illegal water abstraction, illegal charcoal burning, illegal burning of farmlands and grasslands without permission;<br>4.3-Participate in policy establishment and awareness creation at local and national level through discussion forums, engaging political members to lobby for policy reforms in forestry and wildlife sector, strengthening multi-level institutional participation in policy formulation and ensuring local people participate in decision making as stipulated in the policy documents on participatory management of MKFRNP | **4. Monitor and manage, rather than suppress, fires that are of minimal danger to communities, infrastructure or resource values;**<br>4.1-Construct look out towers<br>4.2-Fire monitoring crew/ scouts<br>4.3-Establish access to water sources<br>4.4-Evacuate people<br>**5. Integration of fire management programs aimed at the reduction and control of invasive alien plant species;**<br>5.1-Prevention<br>5.2-Chemical control<br>5.3-Manual control<br>5.4-Cultural control/ competition<br>5.5-Biological control |

**Table A1.** *Cont.*

| Developed Management Strategies (MS) | Management Activities in MKFRNP | Integrated Fire Management (IFM) Activities in MKFRNP |
|---|---|---|
| 5. MS5 Biodiversity conservation interests | 5.1-Ensuring the trees species diversity is conserved and the endangered tree species are protected from illegal loggers through increased patrols, arrest and prosecution of culprits; 5.2-Ensuring the wildlife species diversity is conserved and the endangered wildlife (Rhinos, elephants) are protected from poachers through increased patrols, arrest and prosecution of culprits; 5.3-Key wildlife habitats are protected from human destructive human activities through increased patrols, arrest and prosecution of culprits involved in destruction of key wildlife habitats through illegal grazing, illegal farming, illegal charcoal burning, illegal timber logging and illegal wildlife hunting; 5.4-Reduction and control of invasive alien plant and animal species through prevention, chemical control, manual control, cultural control or competition and biological control in MKFRNP | **6. Minimize the potential occurrence of ecological undesirable fires in ecosystems that have hydrophobic soils;** 6.1-Protection plans 6.2-Protection maps 6.3-Prevention of erosion 6.4-Prevention of loss of organic-rich soils **7. Incorporate land use, forest resource, catchment area and community planning in IFM activities at all appropriate scales;** 7.1-Land use planning 7.2-Forest resource management planning 7.3-Community participation in fire management 7.4-Laws, policy and institutional framework **8. Develop a high level of public awareness and support for IFM;** 8.1-Public meetings 8.2-Posters 8.3-Sign boards 8.4-Radio 8.5-TV 8.6-Newspapers 8.7-Internet 8.8-Social groups **9. Incorporate traditional fire use and management practices when developing and implementing of IFM strategies;** 9.1-Clearing land for agricultural fields 9.2-Replenishing soil nutrients in agricultural fields 9.3-Killing woody species in rangelands 9.4-Encouraging grass growth 9.5-Increasing wild seed production 9.6-Honey collection 9.7-Hunting **10. Reducing IFM costs** 10.1-Staff salaries 10.2-Equipment purchase 10.3-Repair and maintenance 10.4-Fuel costs |
| 6. MS6 Timber production interests | 6.1-Improving the quality of timber produced by thinning and pruning, establishing more timber plantations in deforested areas through PELIS; 6.2-Ensuring timber resource inventories are conducted and timber logs are well priced; 6.3-Managing of wildfires in timber plantations by reducing fuel loads through firewood collection, cutting grass to feed livestock and controlling the use of fire by farmers during land preparation; 6.4-Protect timber plantations from illegal loggers through increased patrols, arrest and prosecution of culprits; 6.5-Reducing game damage on timber plantations through installation of electric fences; 6.6-Ensuring fare allocation of CFAs the harvested areas for plantation establishment and livelihood improvement scheme (PELIS) by considering their registration and participation in management activities | |
| 7. MS7 Community interests | 7.1-Enhancing CFA members participation in fire management by providing training in fire monitoring and firefighting; 7.2-Training CFA members to improve farming of crops and planting of trees under (PELIS), beekeeping in forest to improve their livelihoods; 7.3-Training and supervising CFAs on when to use fire during the farming (PELIS) and honey collection activities within the forest to minimize cases of uncontrolled wildfires; 7.4- Enhancing formation of more CFAs so that they can participate in programmes aimed at encouraging use of fuel efficient wood stoves, payment of firewood collection revenue, payment of livestock grazing revenue, payment of herbal medicine collection revenue, 7.5- Encouraging communities to participate monitoring and reporting of timber and wildlife poaching activities by giving them jobs or incentives; 7.6-Enhancing community user rights through establishment of regulations on who has the right to access and use certain resources and to what extent they can use them without depleting or degrading the resources; 7.7-Encouraging community participation in decision making through open discussion forums, FGDs, voting and voicing of their concerns over certain management decisions that contradict their interests; 7.8-Allowing firewood collection, cutting of grass to feed livestock and livestock grazing to reduce fuel loads in MKFRNP | |

## Appendix B. The Current Management Objectives in the Seven Forest Stations in MKFRNP

**Table A2.** The current management objectives that are being implemented by the KFS and KWS and benefits obtained by various stakeholders in the seven forest stations in MKFRNP.

| Current Management Objectives | Benefits Obtained by Various Stakeholders in MKFRNP | Forest Stations According to Their Fire Danger | | | | | | |
|---|---|---|---|---|---|---|---|---|
| | | Very High Fire Danger | | | High Fire Danger | Moderate Fire Danger | Low Fire Danger | |
| | | Marania | Ontulili | Gathiuru | Nanyuki | Naru moru | Hombe | Chehe |
| 1. Wood production | Timber | Y | Y | Y | Y | Y | Y | N |
| | Poles for building | Y | Y | Y | Y | Y | Y | Y |
| 2. Energy | Firewood | Y | Y | Y | Y | Y | Y | Y |
| | Charcoal | N | N | N | N | N | N | N |
| 3. Biodiversity conservation | Maintaining wildlife species diversity | Y | Y | Y | Y | Y | Y | Y |
| | Maintaining tree species & ground vegetation diversity | Y | Y | Y | Y | Y | Y | Y |
| | Key habitat and protected areas | Y | Y | Y | Y | Y | Y | Y |
| 4. Social values | Religious and cultural sites | Y | Y | Y | Y | Y | Y | Y |
| | Education and research | Y | Y | Y | Y | Y | Y | Y |
| 5. Income | Employment in forestry sector | Y | Y | Y | Y | Y | Y | Y |
| | Tourism | Y | Y | Y | Y | Y | Y | N |
| 6. Agriculture | Livestock grazing | Y | Y | Y | Y | Y | Y | Y |
| | Farming of crops (PELIS) | Y | Y | Y | Y | Y | Y | N |
| | Foraging of wild fruits | Y | Y | Y | Y | Y | Y | Y |
| 7. Non timber forest products | Honey collection | Y | Y | Y | Y | Y | Y | Y |
| | Fishing | N | N | Y | Y | Y | N | N |
| | Hunting of game meat | N | N | N | N | N | N | N |
| | Medicinal plants and spices | Y | Y | Y | Y | Y | Y | Y |
| 8. Protection, water and climate change amelioration | Prevention of soil erosion and landslides | Y | Y | Y | Y | Y | Y | Y |
| | Water quality and quantity | Y | Y | Y | Y | Y | Y | Y |
| | Carbon sequestration | Y | Y | Y | Y | Y | Y | Y |

Where Y = Yes and N = No.

## Appendix C. Priorities of the Criteria after the PWC Using AHP

**Table A3.** Priorities of the criteria in the seven forest stations according to the four fire danger categories after pairwise comparison (PWC) by AHP.

| Objectives | Criteria | Null | Very High Fire Danger | High Fire Danger | Moderate Fire Danger | Low Fire Danger |
|---|---|---|---|---|---|---|
| | | Priorities | Priorities | Priorities | Priorities | Priorities |
| 1. Wood production | 1.1 Timber | 0.500 | 0.750 | 0.667 | 0.667 | 0.667 |
| | 1.2 Poles | 0.500 | 0.250 | 0.333 | 0.333 | 0.333 |
| 2. Energy | 2.1 Firewood | 0.500 | 0.833 | 0.833 | 0.833 | 0.833 |
| | 2.2 Charcoal | 0.500 | 0.167 | 0.167 | 0.167 | 0.167 |
| 3. Biodiversity conservation | 3.1 Maintaining of wildlife species | 0.333 | 0.327 | 0.311 | 0.327 | 0.311 |
| | 3.2 Tree species and ground vegetation diversity | 0.333 | 0.413 | 0.493 | 0.413 | 0.493 |
| | 3.3 Key habitat and protected areas | 0.333 | 0.260 | 0.196 | 0.260 | 0.196 |
| 4. Social values | 4.1 Religious and cultural sites | 0.500 | 0.500 | 0.667 | 0.667 | 0.667 |
| | 4.2 Education and research | 0.500 | 0.500 | 0.333 | 0.333 | 0.333 |
| 5. Income | 5.1 Employment in forestry sector | 0.500 | 0.667 | 0.667 | 0.667 | 0.667 |
| | 5.2 Tourism | 0.500 | 0.333 | 0.333 | 0.333 | 0.333 |
| 6. Agriculture | 6.1 Farming of crops (PELIS) | 0.500 | 0.667 | 0.333 | 0.333 | 0.333 |
| | 6.2 Livestock grazing | 0.500 | 0.333 | 0.667 | 0.667 | 0.667 |
| 7. Non timber forest products | 7.1 Foraging wild fruits | 0.200 | 0.221 | 0.216 | 0.242 | 0.216 |
| | 7.2 Honey collection | 0.200 | 0.342 | 0.360 | 0.242 | 0.360 |
| | 7.3 Fishing | 0.200 | 0.135 | 0.097 | 0.242 | 0.097 |
| | 7.4 Game meat | 0.200 | 0.081 | 0.090 | 0.030 | 0.090 |
| | 7.5 Medicinal plants and spices | 0.200 | 0.221 | 0.236 | 0.242 | 0.236 |
| 8. Protection and climate change amelioration | 8.1 Soil erosion and landslides | 0.333 | 0.327 | 0.327 | 0.311 | 0.327 |
| | 8.2 Water quality and quantity | 0.333 | 0.413 | 0.413 | 0.493 | 0.413 |
| | 8.3 Carbon sequestration | 0.333 | 0.260 | 0.260 | 0.196 | 0.260 |

Firewood, timber, livestock grazing and employment in the forest sector are highly preferred criteria in the seven forest stations after PWC by AHP.

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
