# Peer review of "Evaluating Management Strategies for Mount Kenya Forest Reserve and National Park to Reduce Fire Danger and Address Interests of Various Stakeholders"

_forests, doi:10.3390/f10050426_

Round 1

Reviewer 1 Report

Review of MS 490681: Evaluating management strategies for Mount Kenya  Forest Reserve and National Park to reduce fire  danger and address interests of various stakeholders

I have read this manuscript with interest, as I believe it is important to understand how management strategies is adopted, perceived, and what its impacts are on reducing the fire.

Mount Kenya  Forest Reserve and National Park is a particularly relevant to take on a study like this.

The authors have conducted an extensive process to develop the strategies, with the suport of comunities and other parties. This process can be a good example of good practice for protected area. The process of involving the local communities in developing the management strategies is very important especially to balance the needs and human pressures with biodiversity objectives. I think that the research presented in this manuscript is important and valid.

Comments

Introduction

Some of the information must be clarify and presented:

Is interesting to present what is the cause of the wildfire in the protected area. Is only related to human activity? What activity? Some land burning practices? Tourism? Vegetations?

What are the dimension of these wildfire? Is some historical data related to wildfire in the protected area? What are the human activities allowed in these protected area?

This is the part of the management plan development for this protected area? What is the legal framework of management of protected areas in Kenya? For this process is involve the scientifics for some assessment (e.g biodiversity, species, plants, threats and pressures, anthropic impacts etc.)

These Park have an approval management plan? All the strategies are mentioned in the management plan and must be equally implemented in the protected area? What is the meaning of select? Who implement the management plan (custodians, NGOS, authorities)?

In conclusions, a subchapter about legal framework is need.

2. Material and methods

Very detailed chapter.

Line 95: format of coordinates (degrees and minutes).

Line97-120: the descriptions of site is important. Some sentence have no references (e.g. line 102-103, 117-119 etc.). Do you have some information on vegetations? Is some influence on fire? The elephant corridor is important for management plan?

Some information about tradition on this mountain will be useful. Some information about population is necessary (density, activities etc.).

What are the strategies mentioned in the management plan related to chaparral zone or afro-alpine flora? (That older stands of chaparral become "senescent", thus implying that fire is necessary for the plants to remain healthy; Wildfire suppression policies have allowed dead chaparral to accumulate unnaturally, creating ample fuel for large fires?).

Figure 1: is own elaboration or not? Please indicate the source.

Figure 2: must be done to a high resolution. Also, you can indicate the number of each phase.

Feasibility study. For me is not clear what mean this concept and why is use here. Feasibility study= An analysis and evaluation of a proposed project to determine if it (1) is technically feasible, (2) is feasible within the estimated cost, and (3) will be profitable. Also called feasibility analysis. Maybe the name of these phase must be changed.

What was the process of selection of 21 CFA members at local level? Is representative for communities?

What mean forest stations? What is total number of forest stations in the park? An statistics on wild fire that support the choise of seven station is necessary.

Line 194-195: …selecting the best management strategy. (for wildfire or other? Or is in general?). Create confusion.

Table 1-must be included in text (not on a single page). Eliminate page break.Also table 2 can be move on annexes (strategies). Also, the scores represent the mean? (or number of observation). A legend will be useful.

In this table is a discrepancy. At the begginngs the authors mention that the poverty is a pressure. Why you consider that the participants evaluate the non timber forest products with low values (e.g. hunting or fishing)?

Table 3: is different to follow, try to put on a page.

Line 246: Expert Choice Software. The reference is missing.

3. Results

This chapter is very poor and must be developed.Some statistics (e.g. correlation)  will improve the paper.

Table 5. This is the result of application of Expert Choice software?

4. Discussion

Line 346-350. Very interesting. Also, do you consider that this strategies is applicable on the entire area of Mt. Kenya?

How the best strategies can be implemented in the practices? The paper present the process of develop and choise the strategies but how can be transpose in management plans? MS5 is the best?

What are the improvement of new strategies vs. old management objectives (presented in Table 8-App.A).

I hope that the authors will find my comments useful, and that they will be able to improve this manuscript, as I believe that the material presented is relevant to sustainable management of protected areas.

Author Response

Response to comments from Reviewer I

Dear Reviewer 1: We have managed to respond to all your suggestions, questions and observations, see below.

Introduction

Some of the information must be clarify and presented:

Is interesting to present what is the cause of the wildfire in the protected area. Is only related to human activity? What activity? Some land burning practices? Tourism? Vegetation?

Response: Thank you for the questions

We addressed your questions as shown in a paragraph with lines 49-76

What are the dimensions of these wildfires? Is some historical data related to wildfire in the protected area?

Response: Thank you for the questions

We addressed your questions as shown in a paragraph with lines 77-84

What are the human activities allowed in these protected areas?

Response: Thank you for the questions

We addressed your questions as shown in a paragraph with lines 85-95

This is the part of the management plan development for this protected area? What is the legal framework of management of protected areas in Kenya?

Response: Thank you for the questions

We addressed your questions as shown in lines 96-106

For this process involve the Scientifics for some assessment (e.g biodiversity, species, plants, threats and pressures, anthropic impacts etc.)

Response: Thank you for your suggestion

We addressed your suggestion as shown in a paragraph with lines 107-144

These Park have an approval management plan? All the strategies are mentioned in the management plan and must be equally implemented in the protected area?

Response: Thank you for the questions

We addressed your questions as shown in a paragraph with lines 145-154

What is the meaning of select?

Response: Thank you for the question

We addressed your question as shown in lines 155-160

Who implement the management plan (custodians, NGOS, authorities)?

Response: Thank you for the question

We addressed your question as shown in lines 172-174

In conclusions, a subchapter about legal framework is need.

Response: Thank you for your suggestion

We addressed your suggestion as shown in a paragraph with lines 616-624

2. Material and methods

Very detailed chapter.

Response: Thank you for the compliment.

Line 95: format of coordinates (degrees and minutes).

Response: Thank you for your suggestion

We addressed your suggestion as shown in line 184-185

Line97-120: the descriptions of site are important. Some sentences have no references (e.g. line 102-103, 117-119 etc.).

Response: Thank you for the questions

We addressed your questions as shown in lines 192-194, 208 and 210

Do you have some information on vegetation?

Response: Thank you for the questions

We addressed your questions as shown in a paragraph with lines 211-235

Is some influence of vegetation on fire?

Response: Thank you for the questions

We addressed your questions as shown in two paragraphs with lines 258-268 and another paragraph with lines 269-280

The elephant corridor is important for management plan?

Response: Thank you for the questions

We addressed your questions as shown in lines 136-139

Some information about tradition on this mountain will be useful. Some information about population is necessary (density, activities etc.).

Response: Thank you for the suggestions

We addressed your suggestions as shown in lines 236-244 on traditions and lines 245-257 on population and human activities.

What are the strategies mentioned in the management plan related to chaparral zone or afro-alpine flora? (That older stands of chaparral become "senescent", thus implying that fire is necessary for the plants to remain healthy; Wildfire suppression policies have allowed dead chaparral to accumulate unnaturally, creating ample fuel for large fires?).

Response: Thank you for the questions

We addressed your questions as shown in two paragraphs with lines 258-268 and another paragraph with lines 269-280 that have influence of vegetation on fires

Figure 1: is own elaboration or not? Please indicate the source.

Response: Thank you for the suggestions

We addressed your suggestions as shown in lines 294-295

Figure 2: must be done to a high resolution. Also, you can indicate the number of each phase.

Response: Thank you for the suggestions

We addressed your suggestions as shown in Figure 2 lines 305-307

Feasibility study. For me is not clear what mean this concept and why is use here. Feasibility study= An analysis and evaluation of a proposed project to determine if it (1) is technically feasible, (2) is feasible within the estimated cost, and (3) will be profitable is also called feasibility analysis. Maybe the name of this phase must be changed.

Response: Thank you for the suggestions

We addressed your suggestions as shown in lines 308

What was the process of selection of 21 CFA members at local level? Is representative for communities?

Response: Thank you for the questions

We addressed your questions as shown in lines 339-342

What mean forest stations? What is total number of forest stations in the park? An statistics on wild fire that support the choise of seven station is necessary.

Response: Thank you for the questions and suggestions

We addressed your questions and suggestions as shown in lines 314-323

Line 194-195: …selecting the best management strategy. (for wildfire or other? Or is in general?). Create confusion.

Response: Thank you for the observations

We addressed your observations as shown in lines 377-379

Table 1-must be included in text (not on a single page). Eliminate page break. Also, the scores represent the mean? (or number of observation). A legend will be useful.

Response: Thank you for the questions and suggestions

We addressed your questions and suggestions we eliminated page break in Table 1, added a legend and moved table 1 to results section as suggested by reviewer 2 and is now Table 3 as shown in lines 480-483

Also table 2 can be move on annexes (strategies).

Response: Thank you for the suggestions

We addressed your suggestions and put Table 2 in the Appendix A as Table 4 under strategies as shown in lines 655-657

In this table is a discrepancy. At the beginning the authors mention that the poverty is a pressure. Why you consider that the participants evaluate the non-timber forest products with low values (e.g. hunting or fishing)?

Response: Thank you for the questions and suggestions

We addressed your questions and suggestions in the discussion in lines 593-600.

Table 3: is different to follow, try to put on a page.

Response: Thank you for the suggestions

We addressed your suggestions and put Table 3 on one page and is now Table 1 as shown in lines 459-462

Line 246: Expert Choice Software. The reference is missing.

Response: Thank you for the observations

We addressed your observations as shown in lines 428-429

3. Results

This chapter is very poor and must be developed. Some statistics (e.g. correlation) will improve the paper.

Response: Thank you for your observation and suggestion

We addressed your suggestions to further develop the results chapter and those of reviewer 2. We also decided to move Table 5 in a graphical representation as shown in Figure 4 lines 488-493. We did not see any added value in adding statistics on the descriptive analysis of the management strategies. The AHP allows a cardinal ranking of the alternatives, which allows us to derive the strict preferences directly from the priorities obtained. Therefore we did not include any other statistics.

Table 5. This is the result of application of Expert Choice software?

Response: Thank you for the question

Yes the results of the AHP can be obtained by the application of Expert Choice software as explained in lines 486-491

4. Discussion

Line 346-350. Very interesting, also, do you consider that these strategies are applicable on the entire area of Mt. Kenya?

Response: Thank you for the observation and question

We assume that the results are quite robust. However, we addressed your observation and question as shown in lines 530-533

How the best strategies can be implemented in the practices? The paper present the process of develop and choice the strategies but how can be transpose in management plans? MS5 is the best?

Response: Thank you for the questions

We addressed your questions as shown in lines 569-581

What are the improvement of new strategies vs. old management objectives (presented in Table 8-App.A).

Response: Thank you for the observation and question

We rearranged the Appendix A: Table 8 is now Appendix A: Table 4

We also addressed your observation and question in the discussion as shown in lines 581-587

Thanks in advance,

Kevin Wafula and Harald Vacik,

Reviewer 2 Report

Overall, the authors present a useful documentation of a complex stakeholder engagement process focused on evaluating management strategies in an important ecological and social context. The lessons learned will prove beneficial as others look to reproduce such an approach. There are numerous editorial modifications that should be made to improve clarity of the presentation. I have attached a separate .pdf with handwritten notes and suggested edits. One area for particular improvement is in the presentation of results. Starting at Line 217, much of this text is actually Results vs. Methods and could be moved to the Results section for clarity. Another area that was not clear to me was the extent of the role of community members living adjacent to the MKFRNP. Clearly those with direct interests, such as CFAs, were engaged – but how much involvement in the FGDs did the broader community engage? Other notes: I eventually figured out Table 5, but the values (weights) were not intuitive. Some further explanation of the 0-1 weights would be helpful (or perhaps present a more graphical approach such as Figure 4). Some terminology throughout require some further explanation or modification to more technical terms. These are noted in the scanned .pdf, but include the reference to MKFRNP as a “major water tower” – I think I understand the authors are referring to hydrological benefits of the park, but there may be a more technical way to describe the system. Likewise, other unclear terms used include “excisions” and “abstraction”.

Author Response

Comments of Reviewer II

Dear Reviewer,

We were able to make all the corrections as you had suggested in the PDF text file and in the reviewers comment section. Our paper was also revised by a native English speaker Elaine Siddall who  has a masters degree from the UK in forestry and biodiversity conservation.

Comments and Suggestions for Authors

Overall, the authors present a useful documentation of a complex stakeholder engagement process focused on evaluating management strategies in an important ecological and social context. The lessons learned will prove beneficial as others look to reproduce such an approach. There are numerous editorial modifications that should be made to improve clarity of the presentation. I have attached a separate .pdf with handwritten notes and suggested edits.

Response: Thank you for your positive compliment

One area for particular improvement is in the presentation of results. Starting at Line 217, much of this text is actually Results vs. Methods and could be moved to the Results section for clarity. Response: Thank for your observation and suggestion

We addressed your observation and suggestion and moved Table 1 to results section and is now Table 3 as shown in lines 480-483

Another area that was not clear to me was the extent of the role of community members living adjacent to the MKFRNP. Clearly those with direct interests, such as CFAs, were engaged – but how much involvement in the FGDs did the broader community engage?

Response: Thank you for the questions

We addressed your questions as shown in lines 338-343

Other notes: I eventually figured out Table 5, but the values (weights) were not intuitive. Some further explanation of the 0-1 weights would be helpful (or perhaps present a more graphical approach such as Figure 4).

Response: Thank for your observation and suggestion

We also addressed your suggestions and decided to make Table 5 a graphical representation as shown in Figure 4 lines 491-493

Some terminology throughout require some further explanation or modification to more technical terms. These are noted in the scanned .pdf, but include the reference to MKFRNP as a “major water tower” – I think I understand the authors are referring to hydrological benefits of the park, but there may be a more technical way to describe the system.

Response: Thank for your observation and suggestion

We also addressed your suggestions and decided to change the words

Major water tower to water catchment area

Excisions we change to allocation of forest land to some communities and influential individuals by the former governments

Abstraction we change to water collection.

We also made corrections in the manuscript as you had indicated in the attached PDF file

Thanks in advance,

Kevin Wafula Nyongesa and Harald Vacik

Round 2

Reviewer 1 Report

Congratulation for your work regarding the impovement of the paper. I feel that now, the framework is well detailed. Is more clear to understand the importance of the forest management plan and the involvement of communities in these process.